# Association of plasma metabolites and diagnostic imaging findings with hepatic lipidosis in bearded dragons (*Pogona vitticeps*) and effects of gemfibrozil therapy

Trinita K. Barboza [1¤a]*, Leonardo Susta [2], Alex zur Linden [1‡], Sara Gardhouse [3¤b‡], Hugues Beaufrère [1¤c]*

1 Department of Clinical Studies, Ontario Veterinary College, University of Guelph, Guelph, Ontario, Canada,
2 Department of Pathobiology, Ontario Veterinary College, University of Guelph, Guelph, Ontario, Canada,
3 Health Sciences Center, Ontario Veterinary College, University of Guelph, Guelph, Ontario, Canada

☯ These authors contributed equally to this work.
¤a Current address: Department of Clinical Sciences, Cummings School of Veterinary Medicine, Tufts University, North Grafton, Massachusetts, United States of America
¤b Current address: Department of Clinical Sciences, College of Veterinary Medicine, Kansas State University, Manhattan, Kansas, United States of America
¤c Current address: Department of Veterinary Medicine and Epidemiology, UC Davis School of Veterinary Medicine, Davis, California, United States of America
‡ AL and SG also contributed equally to this work.
* hbeaufrere@ucdavis.edu (HB); Trinita.Barboza@tufts.edu (TKB)

## Abstract

### Objectives

To evaluate the association between plasma metabolites, biochemical analytes, diagnostic imaging findings, and the histologic diagnosis of hepatic lipidosis in bearded dragons. To assess the effects of gemfibrozil therapy on hepatic lipid accumulation and associated diagnostic tests.

### Animals

Fourteen bearded dragons (*Pogona vitticeps*) with varying severity of hepatic lipid accumulation (with and without hepatic lipidosis) were included.

### Procedures

Animals underwent coelomic ultrasound, computed tomography (CT) scans, and coelioscopic hepatic biopsies. Clinical pathology tests included lipidologic tests, hepatic biomarkers, and mass spectrometry-based metabolomics. Animals were medicated with gemfibrozil 6mg/kg orally once a day for 2 months in a randomized blinded clinical trial prior to repeating previous diagnostic testing.

### Results

Hounsfield units on CT were negatively associated with increased hepatic vacuolation, while ultrasound and gross evaluation of the liver were not reliable. Beta-hydroxybutyric-

**Data Availability Statement:** Data for the endoscopic and histologic evaluation, diagnostic imaging, and blood biomarkers including targeted

metabolomics, lipoprotein analysis, and biochemistry were published in the public domain in a permanent scientific data repository (Barboza T, 2022, Replication Data for "Association of plasma metabolites and diagnostic imaging findings with hepatic lipidosis in bearded dragons (Pogona vitticeps) and effects of gemfibrozil therapy", https://doi.org/10.5683/SP3/JG84C7, Scholars Portal Dataverse).

**Funding:** University of Guelph - Ontario Veterinary College - Pet Trust Foundation, grant number: 054460. HB, TB, LS, AZ, SG. https://pettrust.uoguelph.ca The funders had no role in the study design, data collection and analysis, decision to publish, or preparation of the manuscript.

**Competing interests:** The authors have declared that no competing interests exist.

acid (BHBA) concentrations were significantly associated with hepatic lipidosis. Metabolomics and lipidomics data found BHBA and succinic acid to be potential biomarkers for diagnosing hepatic lipidosis in bearded dragons. Succinic acid concentrations were significantly lower in the gemfibrozil treatment group. There was a tendency for improvement in the biomarkers and reduced hepatic fat in bearded dragons with hepatic lipidosis when treated with gemfibrozil, though the improvement was not statistically significant.

## Conclusions

These findings provide information on the antemortem assessment of hepatic lipidosis in bearded dragons and paves the way for further research in diagnosis and treatment of this disease.

## Introduction

Hepatic lipidosis, also known as fatty liver disease or steatosis, is a commonly reported necropsy finding in pet bearded dragons (*Pogona vitticeps)* [1]. A prevalence of 27.3% (156/571) has been reported for moderate to severe hepatic lipid changes in bearded dragons presented for necropsy at two North American pathology laboratories [1]. The disease process involves a progressive accumulation of triacylglycerol in the hepatocytes resulting in disruption of microanatomy, which leads to impaired hepatic metabolism and function, as well as dyslipidemia [2–6]. Hepatic lipidosis is also suspected in clinical cases of bearded dragons that present with non-specific clinical signs including anorexia, lethargy, and weight loss; ultimately progressing to liver failure and death [3, 7–9]. Retrospective data has indicated that age (adult) and sex (female) are risk factors for increased grade and class of hepatic lipid changes [1]. Infectious disease and neoplasia were found to be strong negative predictors for hepatic lipid accumulation and changes [1].

The pathogenesis of hepatic lipidosis in bearded dragons is poorly understood. In reptiles, lipogenesis and fat metabolism are limited in the adipose tissue (primarily the coelomic fat pads) and occurs mainly in the liver, which may increase susceptibility to hepatic triacylglycerol accumulation [10]. Hepatic triacylglycerol accumulation can come from different mechanisms such as increased fatty acid uptake (from food or adipose tissue lipolysis), increased *de novo* synthesis (from carbohydrates), decreased fatty acid degradation (from impaired or saturated mitochondrial β-oxidation), or decreased hepatic excretion (to plasma lipoproteins) [11, 12].

Ante-mortem diagnosis of hepatic lipidosis is challenging in bearded dragons as there is no evidence-based recommendation for the precise definition of hepatic lipidosis. Plasma biochemistry values are largely insensitive to screen for this disorder in reptiles and, as a result most animals have an advanced stage of the disease by the time they are diagnosed; this can subsequently result in poor response to treatment [3, 8, 13]. Since blood is a readily accessible tissue, investigating other commonly used lipid biomarkers such as plasma non-esterified fatty acids (NEFA), ketones, triglycerides, lipoproteins [5, 14, 15], and other metabolites at a larger scale through metabolomics/lipidomics [16–20] may help with screening, diagnosing, and monitoring this disease in a non-invasive manner. In particular, triglycerides would be of interest as hypertriglyceridemia has been demonstrated in red-footed tortoises (*Chelonoidis carbonaria)* with confirmed hepatic lipidosis [5].

Metabolomics is the comprehensive analysis of plasma low molecular weight metabolites, including lipids, by mass spectrometry or nuclear magnetic resonance spectroscopy [21]. It is

revolutionizing the way metabolite disorders are understood, investigated, and diagnosed in mammals and birds [21–24]. Metabolomic profiling of bearded dragon plasma may allow for the identification of key biomarkers associated with various stages of hepatic lipidosis when compared to individuals without significant lipid accumulation. This may help in understanding the affected metabolic pathways in hepatic lipidosis, discover the role of specific metabolites in its pathophysiology, and aid in assessing prognosis and monitoring of treatment [21, 22].

Most of the studies published on lipid metabolism alteration in reptiles have focused on total fatty acid composition of organs and plasma, but not on the lipid molecules or other important lipid classes such as other fatty acyls, glycerolipids, phosphoglycerolipids, sphingolipids, and sterols [25–27]. These studies have demonstrated that several factors can affect the plasma and organ proportion and concentration of lipid [25–27]. Diet, season, body size, and reproductive status can all make interpretation of plasma lipid concentrations difficult as plasma triglycerides are often elevated during vitellogenesis, follicular stasis, prebrumation, and hepatic lipidosis [3, 5, 8, 10, 28].

In the absence of reliable blood biomarkers, the diagnosis of hepatic lipidosis heavily relies on advanced imaging modalities such as computed tomography (CT) and magnetic resonance imaging (MRI), or liver biopsies with histology [3, 8, 13, 29]. To date, histology is the only validated diagnostic tool for hepatic lipidosis [3, 29], and our group has recently proposed a grading system to assess the accumulation of fat and its effects on the hepatic tissue of bearded dragons [30]. However, due to the invasive nature, morbidity and mortality associated with sampling, and cost, histology is typically performed at a later stage in the disease in both human and veterinary medicine and so various imaging modalities continue to be investigated for earlier, non-invasive diagnosis [3, 31–34].

Computed tomography allows for quantitative measurement of radiodensity [Hounsfield Units (HU)] based on the x-ray absorption of various tissues [35]. A decrease in attenuation has been correlated with an increase in hepatic fat content [35]. No study has correlated liver density values on CT images [in HU] with the degree of histologic hepatic fat in bearded dragons. This information may allow for the non-invasive diagnosis and monitoring of hepatic lipidosis as well as quantification of the degree of lipidosis [36]. Ultrasonography of the liver is used in the diagnosis of hepatic lipidosis in mammals [37]. However, this modality lacks specificity and thus has not been proven as a screening tool in human or veterinary medicine [31, 38, 39]. In addition, it only provides a qualitative assessment of hepatic echogenicity and is prone to operator variability [40]. The liver of healthy bearded dragons can have variable echogenicity on ultrasound [41, 42] and the subjectivity of interpretation does not allow a confident diagnosis, especially when compared to more objective CT and histology findings.

Currently, with husbandry changes and supportive care including nutritional support, reversal of hepatic lipidosis could take several months to several years [3]. As in other reptile species, evidence-based therapeutic strategies are lacking for bearded dragons with hepatic lipidosis, as no studies are available. Medical therapies including L-carnitine, methionine, and vitamins have not been proven to be efficacious in reptiles in preliminary trials [9, 13, 43, 44]. Though many of these recommendations are made to increase the catabolism of fat, none are proven to resolve hepatic lipidosis, and discovery of a treatment has the potential to improve the prognosis of this disease. On top of dietary and husbandry changes, a lipid-lowering drug may expedite recovery, reduce hepatic fat, reverse associated dyslipidemia, and aid in the treatment of hepatic lipidosis in bearded dragons. Fibrates are peroxisome proliferator receptor-α agonists that reduce triglyceride levels by increasing fatty acid oxidation in the liver [45–48]. Since triacylglycerols accumulate within hepatocytes in hepatic lipidosis, fibrates such as gemfibrozil have been investigated in mammals and birds [11, 45, 49–51]. In a systematic review of

different fibrates, gemfibrozil was proven to be the most efficacious fibrate in people [46, 52]. The use of fibrates has not previously been studied in reptiles. However, due to the physiological similarities in hepatic lipogenesis and folliculogenesis to birds [53], avian studies were considered to be a good starting point for extrapolation to reptiles.

Despite the common diagnosis of hepatic lipidosis in bearded dragons [1], there is a large knowledge gap regarding its pathophysiology, diagnostic testing, biomarkers, and treatment. The objectives of this study were to: better understand the metabolic derangement associated with hepatic lipidosis in a cohort of bearded dragons with spontaneous disease; to identify potential novel plasma biomarkers that could be used for non-invasive diagnosis and monitoring of hepatic lipidosis using quantitative (targeted) metabolomics and lipidomics; to validate CT as a non-invasive diagnostic tool for hepatic lipidosis in bearded dragons; and to assess gemfibrozil's efficacy on improving liver density on imaging, blood biomarkers, and hepatic fat content on histology. Our hypotheses were that metabolomics, lipidomics, and CT would have significant biomarker differences between bearded dragons with and without spontaneous disease and that gemfibrozil would lead to hepatic fat reduction and biomarker improvement in comparison to a control group of bearded dragons.

## Materials and methods

An animal use protocol was approved for this research by the University of Guelph- Animal Care Committee (Animal utilization protocol #4149). The study was carried out in strict accordance with animal care and use guidelines.

### Inclusion criteria, animals, exclusion criteria

The inclusion criteria for this study was adult bearded dragons of male or female sex that were eating and bright (clinically normal). The study population included 14 bearded dragons (6 males, 8 females) between 1–4 years of age. The number of animals included in this study was based on a sample size estimation using statistical software (G*Power 3.0 [Universität Kiel]) [54] to detect a 10% reduction in hepatic lipid content following gemfibrozil use with a power of 80% and a standard deviation of 5–10% of hepatic lipid. Five sub-adult (1 year old) bearded dragons were acquired through a reptile importer (National Reptile Supply, Mississauga, ON, Canada), and nine adults were acquired from a cohort of relinquished reptiles held in a private herpetological institution (Reptilia Zoo, Vaughan, ON; Ontario Veterinary College, Guelph, ON, Canada). All bearded dragons were considered to be in acceptable health for the duration of the study with good to increased body condition. Any animal with concurrent disease based on physical examination, biochemistry testing, and histology of the liver was excluded from the study.

The bearded dragons were individually housed at the University of Guelph–Central Animal Facility in opaque enclosures that were modified to have ventilation holes and lids half covered with wire mesh. A basking and UVB light (Sunray 70W, Exoterra), polypropylene hide box, and two small bowls for food and water were provided in each enclosure. Each enclosure was lined with newsprint paper sheets (Uline) and were changed when soiled. The temperature was measured via a digital thermometer and maintained at 25˚C in the room and 41˚C directly under the basking light. A 12-hour light cycle was set on timers for the basking and room light with the evening ambient temperature maintained at 25˚C. The bearded dragons were provided a combination of greens (spring mix), fresh cut vegetables (various squash and carrots), gut-loaded and pure calcium dusted house crickets (*Acheta domesticus)* from an in-house colony, and herbivore and insectivore pellets (Mazuri) based on diet recommendations in the literature [29, 55–57]. Water was always available. The greens and vegetables were dusted with 1/

8 teaspoon of pure calcium powder (Repti Calcium Without D$_3$, Zoo Med Laboratories; 43% Ca/kg; 43% Ca/kg) when offered and 1/8 teaspoon of multivitamin once a week [58–60]. On alternating days, twice a week, the bearded dragons were offered 6–12 adult gut-loaded crickets (depending on size). Three days a week, the bearded dragons' diet consisted of a combination of two teaspoons of herbivore and one teaspoon of insectivore reptile pellets which were soaked with water to encourage consumption [55]. However, the majority of the animals elected to self-fast when offered pellets alone aside from the year-old bearded dragons.

On a daily basis the bearded dragon food intake, fecal and urate output, and any behavioral or physical changes were logged (e.g. shedding, hiding) to monitor health of the animals. On a weekly basis, the bearded dragons were weighed, soaked in warm tap water for 15 minutes, and had their enclosures sanitized with an accelerated hydrogen peroxide solution. The bearded dragons were acclimatized for 1 month prior to initiation of the diagnostic study.

## Biochemical panel

Biochemical and diagnostic imaging testing was performed on all bearded dragons to further asses their health, obtain values on lipid metabolism and liver analytes, and screen for pre-existing hepatic lipidosis. At the end of the acclimation period, the bearded dragons were fasted for 24 hours and a 0.9 ml to 1.0 ml blood sample was collected from the caudal tail vein or right jugular vein with minimal to no lymph contamination as noted visually. A 25-ga or 26-ga needle and disposable 1-ml syringe were used, with blood divided into two different heparinized tubes without a serum separator (Microtainer, Becton and Dickinson, Mississauga, ON, Canada). The tubes were inverted a minimum of 5 times and placed on ice. Once all samples were collected, the blood was centrifuged for 10 minutes at 1500g.

One heparinized plasma sample was submitted to the laboratory (University of Guelph, Animal Health Laboratory, Guelph, ON, Canada) for a custom-made biochemistry panel using a clinical reference laboratory analyzer (Cobas c501, Roche Diagnostics International AG, Rotkeruz, Switzerland) that included lipids (total cholesterol, triglycerides, high-density lipoprotein (HDL)-cholesterol, total NEFA, bile acids), liver enzymes (asparate aminotransferase [AST], alanine aminotransferease [ALT], gamma-glutamyl transferase [GGT], glutamate dehydrogenase [GLDH]), and metabolites (beta-hydroxybutyric acid (BHBA), glucose).

## Targeted metabolomics

The second heparinized plasma tube had approximately 0.2ml plasma for metabolomics stored into cryopreservation vials and frozen at -80˚C until analysis. Samples were shipped on dry ice to the analytical laboratory (Analytical Facility for Bioactive Molecules, The Hospital for Sick Children, Toronto, Canada). Heparinized plasma samples were analyzed for 630 metabolites using a standardized metabolomics kit (MxP® Quant 500 kit [Biocrates Life Sciences, Innsbruck, Austria]). The panel included 1 alkaloid, 1 amine oxide, 20 amino acids, 30 amino acid related metabolites, 14 bile acids, 9 biogenic amines, total hexoses, 7 carboxylic acids, p-cresol sulfate, 12 non-esterified fatty acids, 4 hormones, 4 indoles and derivatives, hypoxanthine, xanthine, choline, carnitine and 39 acyl-carnitines, 14 lysophosphatidylcholines, 76 phosphatidylcholines, 15 sphingomyelins, 28 ceramides, 8 dihydroceramides, 19 hexosylceramides, 9 dihexosylceramides, 6 trihexosylceramides, 22 cholesteryl esters, 44 diacylglycerols, and 242 triacylglycerols. In addition, the calculated value of total ceramides was obtained by adding all of the individual ceramide concentrations.

To minimize contamination, solvents used were of liquid chromatography- mass spectrometry grade and glassware was triple-rinsed with pure water (Milli-Q® water, MilliporeSigma™), isopropanol and methanol. Samples, standards, and controls (10uL) were added to a

96-well filter plate with internal standard, dried under nitrogen gas, and then extracted with methanol, as per kit instructions. Extracted samples were analysed by liquid chromatography-tandem mass spectrometry using the supplied high performance liquid chromatography column (Agilent 1290 HPLC system, Agilent Technologies: Santa Clara, California, USA) coupled to a mass spectrometer (Sciex Q-Trap 5500, Sciex, Framingham, MA) both in positive and negative polarity. Data were acquired and quantified using two different software programs (Analyst, version 1.6.3, AB Sciex Pte. Ltd, Singapore; MetIDQ, version 9.7.1-DB110-Oxygen 2893, Biocrates Life Sciences, Innsbruck, Austria) [61]. All metabolites were quantified in umol/L.

## Advanced lipoprotein profiling

Approximately 2 months following initial blood collection, the bearded dragons were re-sampled in a similar manner and an additional 0.5 ml of blood was collected and placed in an EDTA tube. Plasma was separated and frozen at -80˚C until shipping. The samples were then shipped on dry ice to a laboratory (Skylight Biotech Inc., Liposearch® panel, Akita, Japan) for advanced lipoprotein profiling. The technique used gel permeation- high performance liquid chromatography to separate lipoproteins by size prior to chemical analyses, as previously described in mammals and birds [24, 62, 63]. Lipoproteins were analyzed for cholesterol concentration, triglycerides concentration, and particle numbers across 4 major classes and 20 sub-fractions. The concentration of free glycerol was also measured. Lipoprotein particle sizes were also obtained for the main classes as calculated from cholesterol plots using a proprietary algorithm [63]. Non-HDL cholesterol was also calculated as an additional lipoprotein biomarker.

## Imaging (computed tomography and ultrasound)

Following blood collection, the bearded dragons were sedated with 10 mg/kg alfaxalone (Alfaxan Multidose, 10mg/mL, Jurox Animal Health, ON, Canada) [64] IM (intramuscular) and 1 mg/kg hydromorphone (Hydromorphone hydrochloride injection, 2mg/mL, SteriMax, Oakville, ON, Canada) [65] IM in the thoracic limbs. The animals were sedated within approximately 10 minutes following drug administration for coelomic non-contrast enhanced CT scans and subsequent coelomic ultrasounds. Each bearded dragon was placed in ventral recumbency for CT image acquisition. The CT was performed using a 16-slice CT scanner (GE Bright Speed, General Electric Healthcare, Milwaukee, WI). Data was reformatted with routine bone and soft tissue algorithms. Slice thickness was 0.625 mm, and images were reformatted into 1.3 mm slices. The field of view was 25 cm, kVP 120, mA 100. The pitch was 0.938:1, with a 1 sec rotation time. Viewing software (AGFA Enterprise Imaging XERO Viewer 8.1.2, Agfa HealthCare N.V., 2017, Belgium) was used to obtain and analyze images through multi-planar reconstruction (MPR) to allow for better visualization of the liver. The hepatic density was measured in HU using a standardized ROI (region of interest) excluding blood vessels by a single observer. Three circular ROIs were selected from a dorsal MPR view from the caudoventral margin of the left hepatic lobe in the region of hepatic biopsies with an approximate area of 0.05+/- 0.001 cm$^2$ for each ROI. The area of ROI was triangulated to the widest part of the liver using the transverse and sagittal planes.

Following CT imaging, coelomic ultrasound (Philips iU22, Philips Ultrasound, Bothell, USA) was performed on the pediatric abdomen setting with a gain of 76%, and maximum depth for each animal. The animals were manually restrained in dorsal recumbency and a C85 transducer was used to acquire images of the left and right liver lobes and coelomic fat pads when visible by a board certified radiologist (AZ). Images were standardized and recorded for

blinded evaluation by a single experienced observer (final year radiology resident) for liver echogenicity (as a score of normal [no-lipid to mild class], hyperechoic but hypoechoic to fat pads [moderate class], and hyperechoic that is isoechoic to the fat pads [severe class]) with fat pads used as a reference organ. The images were also measured as mean pixel intensity in a standardized ROI following 8-bit grey scale conversion using an imaging processing program (ImageJ 1.53a, National Institutes of Health, USA) [66]. A maximum ROI was chosen in each image to include as much of the target organ as possible (left and right liver or fat pad) using free hand selection, excluding large vessels and liver capsule. The mean pixel intensity of each target organ was calculated and the ratio of liver/fat pad pixel intensity was reported to account for dragon-to-dragon variability in echogenicity. Following imaging, subcutaneous fluids (10ml/kg) were given to each bearded dragon prior to returning them to their enclosures for recovery.

## Coelioscopic guided liver biopsy and histology

Following imaging, the bearded dragons were re-sedated with 10 mg/kg alfaxalone and 1 mg/kg hydromorphone IM, and a coelioscopic guided liver biopsy was performed under isoflurane (IsoFlo, Abbott Laboratories, North Chicago, IL, USA) general anesthesia with a 2.7mm rigid endoscope (Karl Storz Endoscopy Canada Ltd., Mississauga, ON, Canada) and biopsy forceps using a paramedian approach with $CO_2$ insufflation as described in reptiles [67]. All efforts were made to minimize suffering including minimally invasive biopsies and 3 days of post-operative analgesia (tramadol, 10 mg/kg PO q24h, compounded product, Ontario Veterinary College, Guelph, ON, CAN). As all bearded dragons were not sampled on the same day, they were selected for the procedure in a random sequence by mixing name cards and selecting a random card. Images of the liver were captured for each bearded dragon, and subsequently 5-French endoscopic biopsy forceps were used for liver sampling. Coelioscopic images were subjectively graded on an ordinal scale by a blinded observer (SG) based on color, parenchyma texture, margination, and size. Categories included no-lipid (brown, smooth, sharp margins), mild (yellow-brown, sharp margins), moderate (yellow, rounded margins), and severe (yellow, possible capsular fibrosis, rounded margins, large) lipidosis.

Coelioscopic biopsies were fixed in 10% buffered formalin and submitted to Animal Health Laboratory for processing where they were embedded in paraffin, sectioned onto glass slides, and stained with hematoxylin and eosin. Additional stains including Periodic acid-Schiff, Masson's trichome, and Congo Red were utilized to further evaluate cases as needed. The grade and classification of hepatic lipid changes were determined for each histological sample according to a scoring system which assessed percent of hepatocellular vacuolation, fibrosis, and hepatocellular swelling by a blinded pathologist (LS) and an exotic animal veterinarian (TB), as previously described S1 Table [30]. In addition, the percentage of hepatic lipid was calculated using digital image analysis with an image processing program (ImageJ 1.53a) for each animal [66]. Due to the physiological process of lipogenesis in the bearded dragon liver [10], and the histological evaluation of 252 cases [30], authors suspected that all vacuolated hepatocytes without cellular swelling were a variation of normal. Therefore, based on the grading system where cellular swelling placed cases in the moderate or severe class [30], cases with moderate to severe hepatic lipid changes were considered most likely to be diagnosed with hepatic lipidosis.

A male bearded dragon who was eating and defecating well postoperatively was found deceased 14 days after his coelomic biopsies from septicemia with coelomitis. Evaluation of the postmortem histology slides indicated the presence of hepatic granulomas, suggesting bacterial dissemination / sepsis.

Bearded dragons (n = 13) had a mean ± sd weight of 303.3 ± 166 g. Animals in the no-lipid-to-mild class (n = 6) weighed significantly less (p<0.001, t-test) at 150.0 ± 49.8 g compared to animals in the moderate-to-severe class (n = 7) at 434.7 ± 97.2 g. Those in the no-lipid-to-mild class were 1 years of age and smaller in size, except for 1 subject who was 3 years of age and full grown. Bearded dragons in the moderate-to-severe class were between 2–4 years of age and larger in size.

## Gemfibrozil clinical trial

Following a two-month recovery period from liver biopsies, the bearded dragons were randomized, blocking for sex and hepatic lipidosis status (as assessed by histology), using statistical software (R, version 4.0.3, R foundation for statistical computing, Vienna, Austria) [68], into 2 treatment groups of 6 individuals: a control group which received the base used to compound the medication (equivalent volume to mg/kg dose) and a treatment group that received 6mg/kg of gemfibrozil (Teva-Gemfibrozil, 600mg tablet, Teva Canada Limited, Toronto, ON, Canada) orally once a day in the morning. The dose of gemfibrozil was obtained by allometric scaling based on the known safety of the drug in a wide variety of animals [69]. The clinical trial was performed over a period of 2 months, with missed weekends for the first month due to a communication error with staff. Treatment administration was blinded. The gemfibrozil was compounded from commercial tablets into an oral aqueous suspension at 12mg/ml. One tablet was dissolved into 25ml of distilled water and then mixed into 25ml of a suspending vehicle (ORA-Plus, Perrigo, Brooklyn, NY, USA) before each administration. Gemfibrozil has an aqueous solubility up to 10 mg/ml [70] and it readily solubilized at the proposed concentration. Bearded dragons were evaluated daily by a blinded observer for side-effects such as reduced appetite and activity, and their food intake and weight were recorded. Husbandry and diet were continued as previously described and if the animals were noted to be anorexic or losing weight, a physical examination was completed and they were syringe fed once every 3 days. Subcutaneous fluids (10 ml/kg) and enemas (3% of body weight; 50:50 water: lubricating jelly [Muko]) were given on an as needed basis. Bearded dragons would be removed from the gemfibrozil trial if they stopped eating for more than a week, showed gastrointestinal signs such as diarrhea, or if they became lethargic. None of the bearded dragons had long term anorexia during the study.

Following 2 months of treatment, the bearded dragons were fasted for 24 hours prior to blood collection (1.0 ml from the caudal tail vein) for a hepatic biochemistry panel including BHBA and a metabolomics panel in a similar manner to pre-treatment sampling. Computed tomography scans, ultrasound, and coelioscopic guided liver biopsies were also performed in the same manner as the initial diagnostics. Two male and two female bearded dragons were humanely euthanized with 1.0 mL of potassium chloride into the caudal tail vein following final sample collection due to the diagnosis of moderate-to-severe histological changes (hepatic lipidosis). Death was confirmed by cessation of heart sounds on doppler. Coelioscopic biopsies were fixed, submitted, stained, and graded as previously outlined. Computed tomography and ultrasound images were also evaluated as previously described. All data were compared between the 2 groups as well as with their baseline data.

## Statistical analysis

**General.** For the purpose of the analysis, and the authors definition of hepatic lipidosis, the histologic classification of lipid changes was simplified to a binary outcome variable with "no-lipid-to-mild" and "moderate-to-severe" values based on a partially validated histology grading system [30].

Diagnostic imaging data were compared between the 2 simplified classes using t-tests or Mann-Whitney U test if not normally distributed (as determined by Shapiro-Wilk tests and quantile plots) or with a heterogeneous variance. Diagnostic utility, for classification into the two simplified histological classes, was assessed using receiver operating characteristic (ROC) curve analysis with area under the curve (AUC) to determine accuracy (1 being 100% accurate) and optimal cut-off to maximize sensitivity and specificity.

The association between the amount of fat in the liver biopsy assessed by digital image analysis and CT liver density, assessed in HU, was investigated using a linear regression analysis. Assumptions were assessed on residual plots and residual diagnostics.

Agreement between the coelioscopic and ultrasound images of the liver and severity of lipid changes on histology (reduced to two classes) was evaluated by calculating the percentage of agreement.

An alpha of 0.05 was used for statistical significance. Statistical software (R, version 4.0.3) was used for analysis and development of graphs [68].

**Biochemistry and metabolomics data.** Metabolites quantified by mass spectrometry, analytes quantified by the clinical analyzer, and lipoprotein analytes on the advanced lipoprotein panels were analyzed together with multivariate statistics. The statistical strategy followed a standard statistical workflow recommended in metabolomics data [71].

Data were first filtered to remove non-informative variables and to increase the statistical power. Variables with a constant or single value across all samples, variables with more than 2 missing values, and variables with values above or below the quantification limits were removed. A total of 238/687 (34.6%) of variables were retained for multivariate analysis. Data were then log transformed and mean centered for variables to be comparable.

Differences in each analyte between simplified classes were then assessed using serial t-tests with an alpha of 0.05 ($p < 0.05$) and an false discovery rate (FDR) of 0.05 ($q < 0.05$) for significance. Clustering was investigated using principal component analysis to detect metabolomic signatures of the simplified hepatic lipid change classification using an unsupervised technique. Clustering was further investigated using a supervised classifying multivariate tool for high-dimensional data: sparse partial least squares–discriminant analysis (sPLS-DA) initially using the 10 variables per component. Loading plots were inspected to detect important metabolites for classification. A heatmap with hierarchical clustering was also generated for data exploration using the 75 most important analytes based on the lowest p-values, as assessed by t-test on normalized data.

A biomarker analysis was also performed. Raw values were used instead of normalized values. Univariate ROC curve analyses were performed and diagnostic accuracy of biomarkers to separate animals in the two simplified histological categories were ranked based on the AUC (1 being 100% accurate). For selected variables, the optimal cut-off to maximize sensitivity and specificity were also obtained. Statistical software (R, version 4.0.3) [68] and metabolomic analysis software (MetaboAnalyst 5.0, Xia Lab, Montreal, QB, Canada) [72] were used for statistical analysis and development of graphs.

**Gemfibrozil trial.** For data obtained after 2 months of gemfibrozil therapy, only specific variables of interest were investigated as directed by prior analysis to increase statistical power. Advanced lipoprotein profiling was not repeated post-gemfibrozil treatment; the effect of gemfibrozil on these variables was thus not investigated. Linear mixed models were performed with the histopathological grade, time (baseline and at completion of treatment), and treatment (control, gemfibrozil). The interactions were treated as fixed effects and individual bearded dragons were treated as the random effect in these models. Assumptions of normality and homoscedasticity were checked on residual plots.

An alpha of 0.05 was used for significance. Statistical software was used for statistical analysis and development of graphs (R, version 4.0.3) [68].

## Results

### Histology of hepatic biopsies

Following evaluation of the histology slides, a female bearded dragon was excluded from the study due to severe hepatic amyloidosis (Congo red stain) and atrophied hepatocytes, without hepatic lipidosis.

A total of 6 males and 7 females met the final inclusion and exclusion criteria for this study. Based on the histological grading system, there were 2 cases in the no-lipid class, 4 in the mild class, 5 in the moderate class, and 2 in the severe class. The scores were aggregated into two categories and out of 13 bearded dragons, 6 dragons (3 males and 3 females) were in the no-lipid-to-mild class on histology and 7 dragons (3 males and 4 females) were in the moderate-to-severe class.

Data for the endoscopic and histologic evaluation, diagnostic imaging, and blood biomarkers including targeted metabolomics, lipoprotein analysis, and biochemistry were published in the public domain in a permanent scientific data repository (Barboza T, 2022, Replication Data for "Association of plasma metabolites and diagnostic imaging findings with hepatic lipidosis in bearded dragons (*Pogona vitticeps*) and effects of gemfibrozil therapy", https://doi.org/10.5683/SP3/JG84C7, Scholars Portal Dataverse).

### Diagnostic imaging

The hepatic density measured by HU was significantly lower in the moderate-to-severe class than the no-lipid-to-mild class in these bearded dragons (p = 0.003, Mann-Whitney U test) (Fig 1). Receiver operating characteristic curve analysis showed an AUC of 0.93 with an optimal cut-off of 21 HU for a sensitivity of 86% and specificity of 100% (Fig 2). The CT HU was linearly associated with hepatic fat content. With each 10% increase in hepatic fat on digital image analysis (ImageJ 1.53a) [66], there was a corresponding decrease of mean ± standard error of the mean of 4 ± 1 in hepatic HU (Fig 3) ($R^2$ = 0.63, p = 0.001).

The diagnostic usefulness of ultrasound was assessed similarly. The hepatic/fat pad echogenicity ratio (determined by pixel intensity) was not significantly different between the 2 classes of histological grading in these bearded dragons (p = 0.063) (Fig 4). The liver had a tendency to increase in echogenicity when compared to the fat pads, but it was not significant. The ROC AUC was 0.69.

There was a 71% agreement between coelioscopic image classification based on visual observation alone and histologic lipid change classification and a 54% agreement between ultrasound images classification and histologic lipid change classification when using the simplified form.

### Biochemistry, lipoprotein profile, and metabolomics

On serial t-tests when adjusting for an FDR of 0.05, the plasma concentration of only 1 metabolite was found to be statistically significant. Beta hydroxybutyric acid was significantly lower in bearded dragons in the moderate-to-severe class (p<0.001, q = 0.037) (Fig 5).

Histologic classes did not cluster well on principal component analysis, but they were well clustered on sPLS-DA (Fig 6). The sPLS-DA model explained 25.9% of the variance. Using loading plots for the first component (Fig 6), the most important discriminating variables between the simplified classes were BHBA and succinic acid. Several other variables also

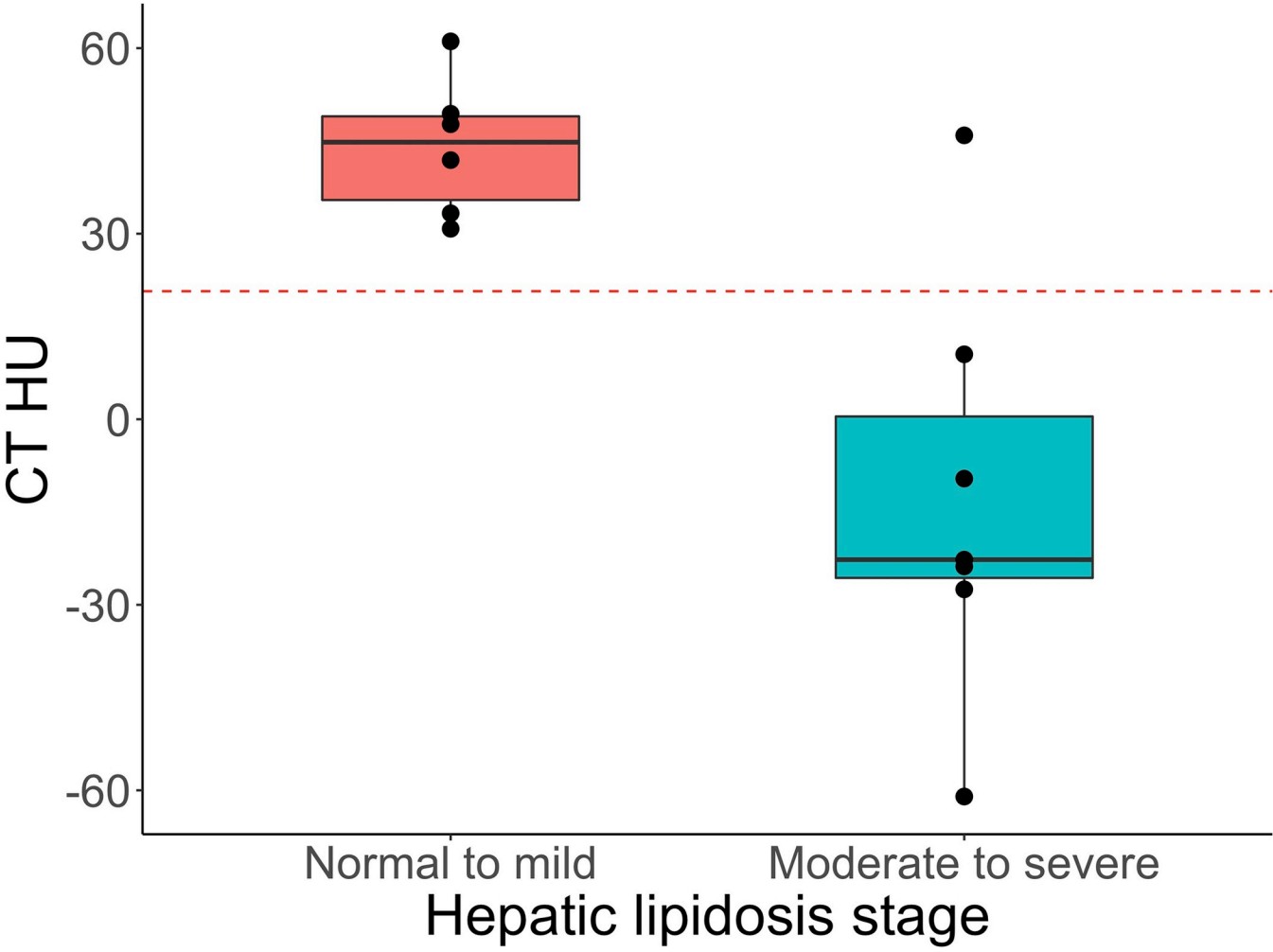

**Fig 1. Computed tomography images (coronal multiplanar reconstruction views) of bearded dragons livers with different histologic classes based on an established grading system [30].** A) no-lipid, 48 Hounsfield Units (HU). B) Mild, 7HU. C) Severe, -36HU.

significantly contributed to the classification, but in a lesser measure and included mainly lipids, especially related to triacylglycerol metabolism such as chylomicron subtypes, chylomicron particle number, large VLDL(very-low density lipoprotein) (VLDL1), TG (triacylglycerol) 16:0_36:2, and PC (phosphatidylcholine) 28:1.

The heatmap also suggested different metabolomic and biochemical signatures between classes (Fig 7). A few metabolites including BHBA, carnitine, lysine and some lipids such as total cholesterol, non-HDL, short-chain acyl-carnitines, NEFA, and others were found to be in lower plasma concentrations in the moderate-to-severe class. Conversely, metabolites dominated by lipids (especially triglycerides in C52, lipoprotein particle numbers and many lipoprotein subtypes) and other metabolites such as succinic acid, free glycerol, lactic acid, and a hepatic marker ALT, were found to be in higher plasma concentrations in the moderate-to-severe class. Fig 8 shows side by side boxplots of metabolites of interest shown to be important in the pathophysiology of hepatic lipidosis in various species [50, 73–86]. These were not statistically significant on serial t-tests.

Regarding biomarker analysis, on univariate ROC curve analysis, BHBA and succinic acid were found to be the best biomarkers with a similar AUC of 0.98. In this cohort of bearded

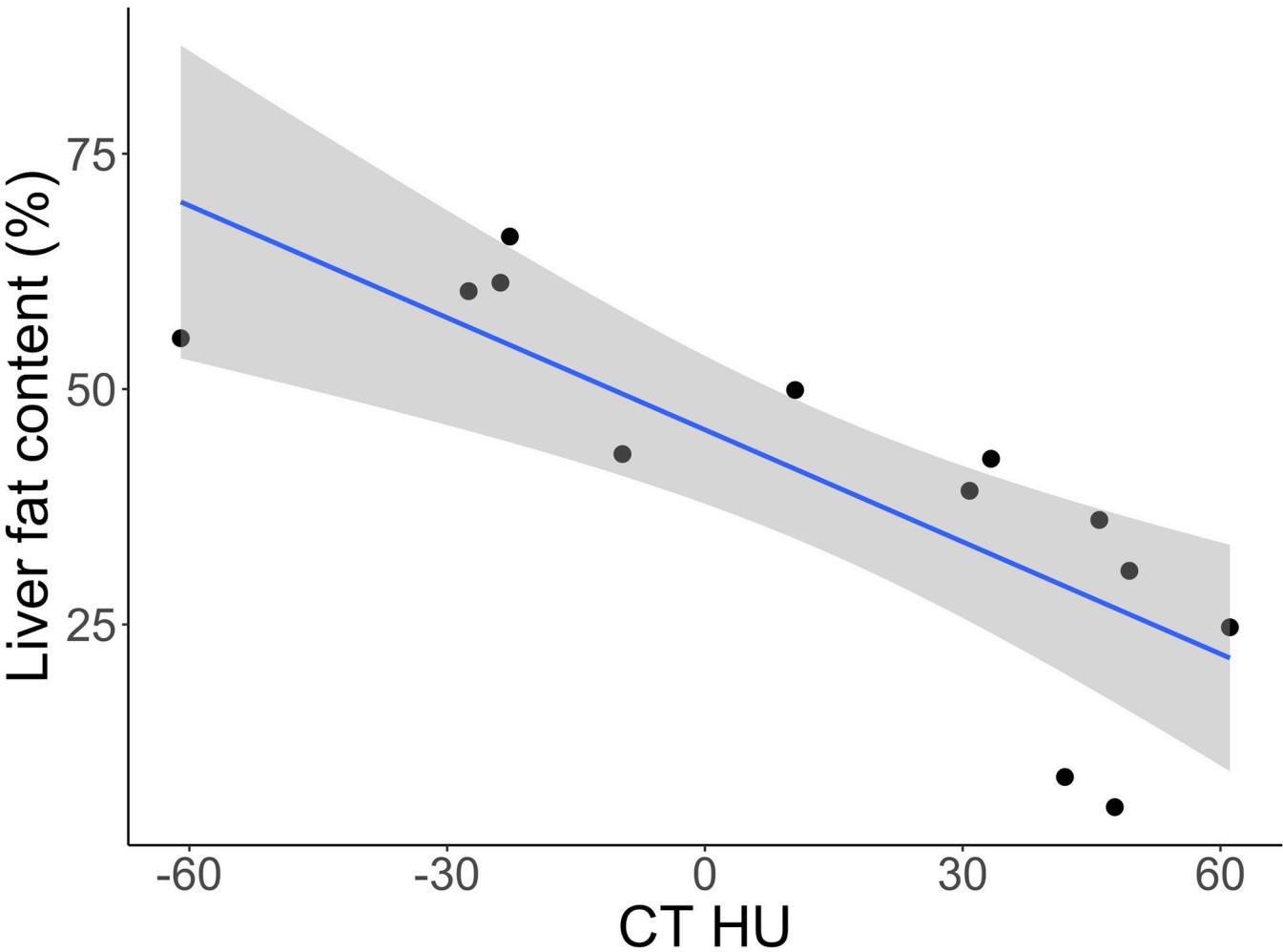

**Fig 2. Box plot with superimposed dot plot of hepatic computed tomography hounsfield units (HU) across two simplified classes of hepatic lipid changes (no-lipid to mild and moderate to severe) based on an established grading system in bearded dragons [30] evaluated by a Mann-Whitney U test.** The dotted red line represents a cut-off value of 21 HU to separate the classes of no-lipid to mild and moderate to severe as determined on receiver operating characteristic curve analysis.

dragons, a BHBA lower than 272 umol/L or a succinic acid higher than 13.7 umol/L gave a sensitivity of 86% and a specificity of 100%.

### Gemfibrozil trial

**Animals.** There were no significant adverse effects such as weight loss, anorexia, or diarrhea during the gemfibrozil trial, though many bearded dragons had fluctuations in their weight throughout the study. One male bearded dragon from the control group was diagnosed with ulcerative cloacitis near the end of the study period and was humanely euthanized following sample collection. This bearded dragon was not excluded from the study as the illness developed at the end of the treatment period.

**Histology of hepatic biopsies.** Among the cases that were classified as no-lipid-to-mild prior to the treatment trial, the control group had two cases that progressed in grade, with one moving from the mild to moderate class, and one case that improved by a single grade. The gemfibrozil group had one case progress from no-lipid to mild and another case that remained unchanged. Among cases that were classified as moderate-to-severe prior to the treatment

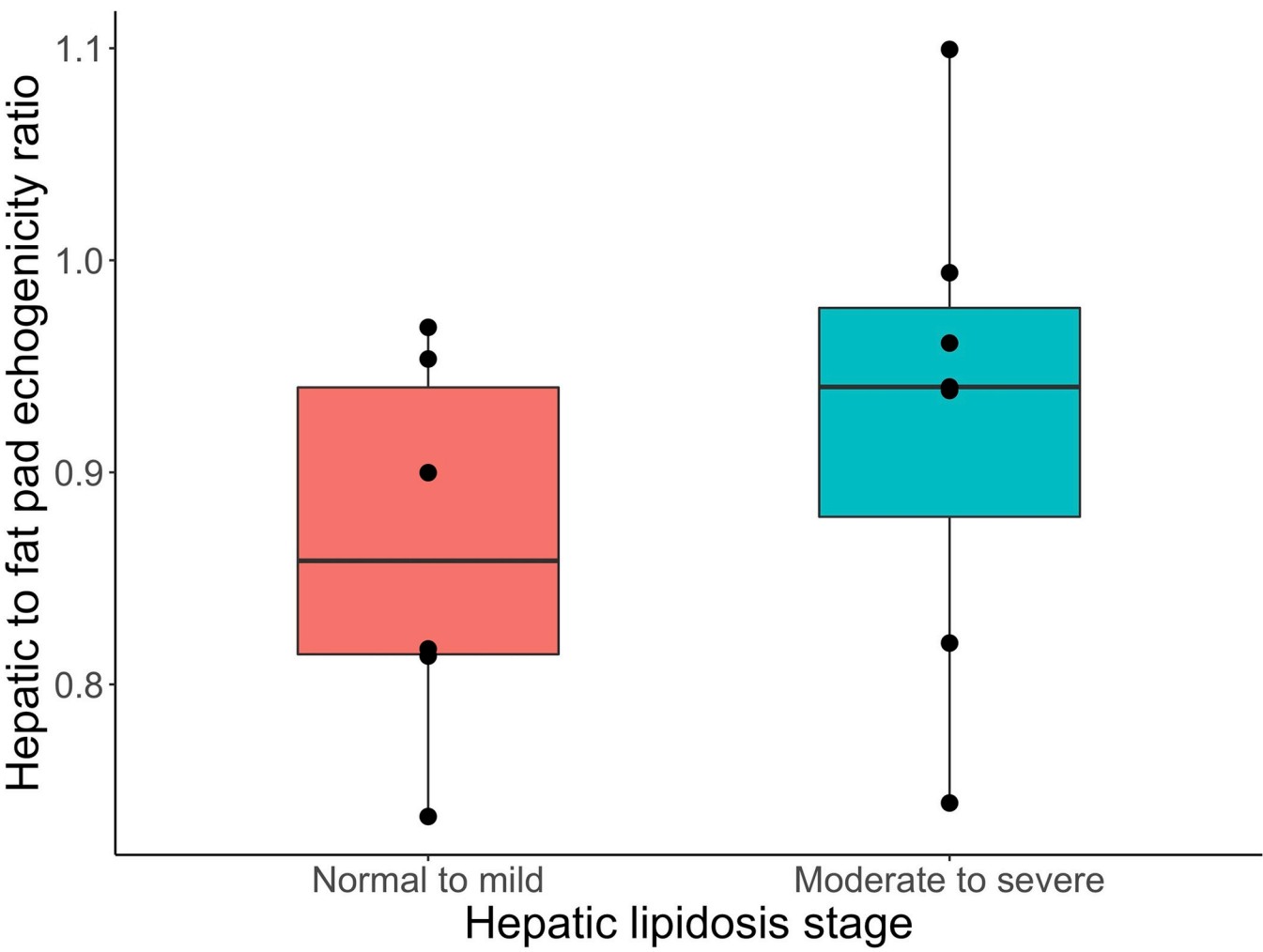

**Fig 3. Scatter plot of bearded dragon liver lipid content (%) from histological samples measured by digital analysis as a function of hepatic density (Hounsfield units) measured on computed tomography and analyzed using linear regression.** The blue line represents the regression line and the shading the 95% confidence interval.

trial, the control group had one case progress by a single grade, while the gemfibrozil group had two cases improve in grade and class, and two cases remained unchanged.

**Efficacy.**   There was a significant effect of gemfibrozil on succinic acid plasma concentrations between treatment groups (p = 0.014). While succinic acid significantly increased over time in the control group (p = 0.015), it did not with gemfibrozil (p = 0.24) controlling for the simplified lesion class. There was no effect of treatment on the other tested outcome variables [BHBA (p = 0.95), triglycerides (p = 0.97), CT HU (p = 0.31), fat percentage on digital image analysis (p = 0.26)] controlling for the simplified lesion class. While effects other than for succinic acid were not significant, trends were seen on graphs. Animals had a tendency to progress in their histological class over time with an associated decrease in BHBA and an increase in triglycerides in the no-lipid-to-mild class (Fig 9). Gemfibrozil did not seem to prevent the progression of hepatic lipid accumulation in this class; however, there was a tendency in animals in the moderate-to-severe class to improve their biomarker profile (increase BHBA, decrease succinic acid, stabilize triglycerides) and decrease the amount of fat in their liver (lower fat percentage on digital image analysis and higher CT HU) (Fig 9).

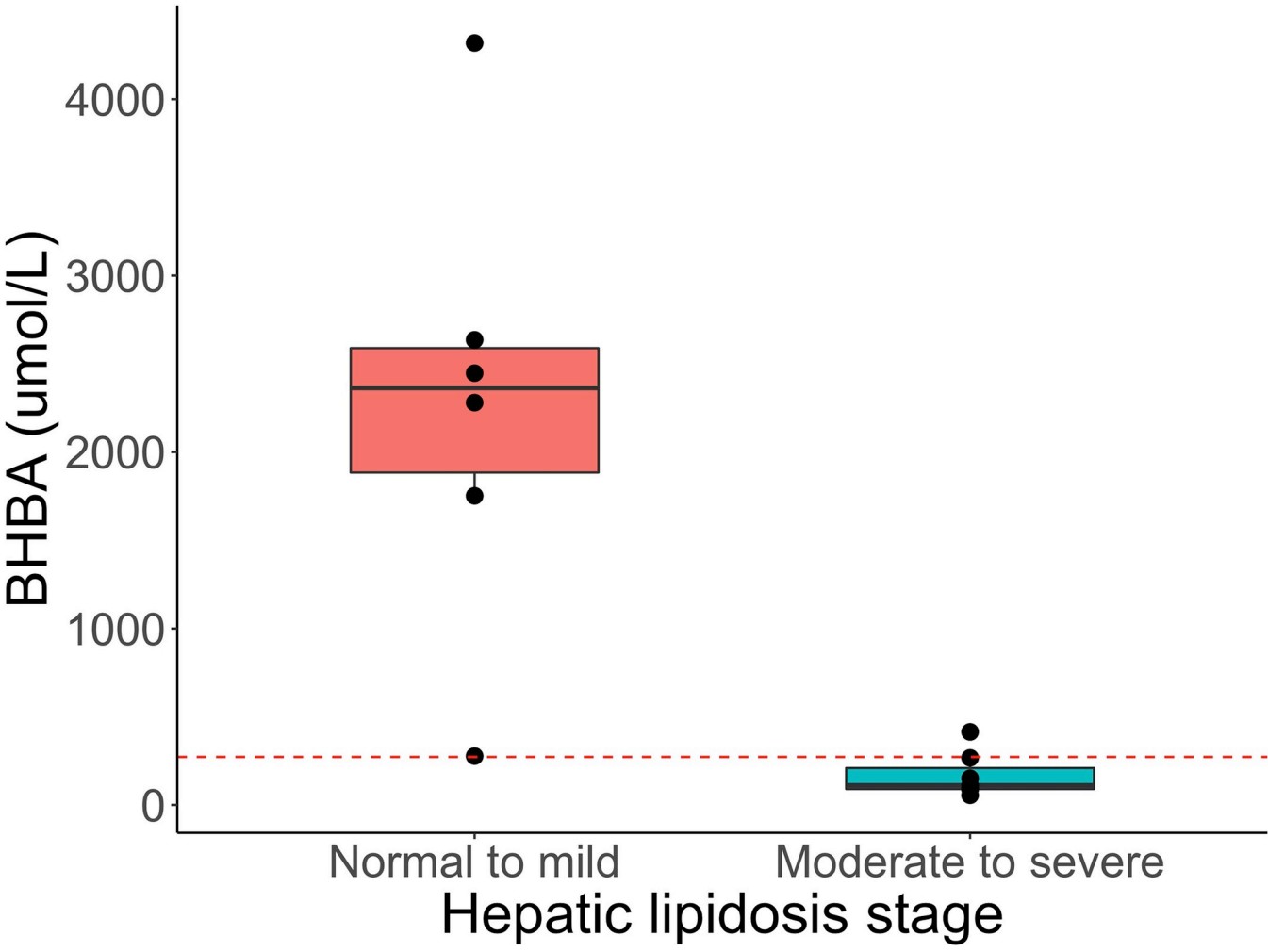

**Fig 4. Box plot with superimposed dot plot of hepatic to fat pad echogenicity ratio on ultrasound across two simplified classes of hepatic lipid changes (no-lipid to mild and moderate to severe) based on an established grading system in bearded dragons [30] evaluated by a t-test.**

## Discussion

This report provides strong evidence for the use of CT in the evaluation of hepatic lipid accumulation in bearded dragons and identifies potential non-invasive biomarkers for hepatic lipidosis. It also provides preliminary data for further investigation of the efficacy of gemfibrozil for the treatment of hepatic lipidosis in this species. CT images were highly accurate (AUC of 0.93 with a HU of 21 or less) in the diagnosis of hepatic lipidosis through the moderate-to-severe class. Hepatic density, based on HU, was also found to be linearly associated with hepatic fat, assessed by image analysis of histology slides; with every 10% increase in fat corresponding to an approximate decrease of 4 HU. Ultrasonography and examination of coelio-scopic images were not found to be sensitive enough to diagnose most cases. Only 1 metabolite, BHBA, was found to be statistically significant in differentiating no-lipid-to-mild class from the moderate-to-severe class; and was significantly lower in bearded dragons with hepatic lipidosis. This biomarker performed better (AUC 0.98) than CT in discriminating the no-lipid-to-mild class from the moderate-to-severe class on fasted bearded dragons, and is an inexpensive test. Succinic acid, another biomarker for hepatic lipidosis, was also found to be

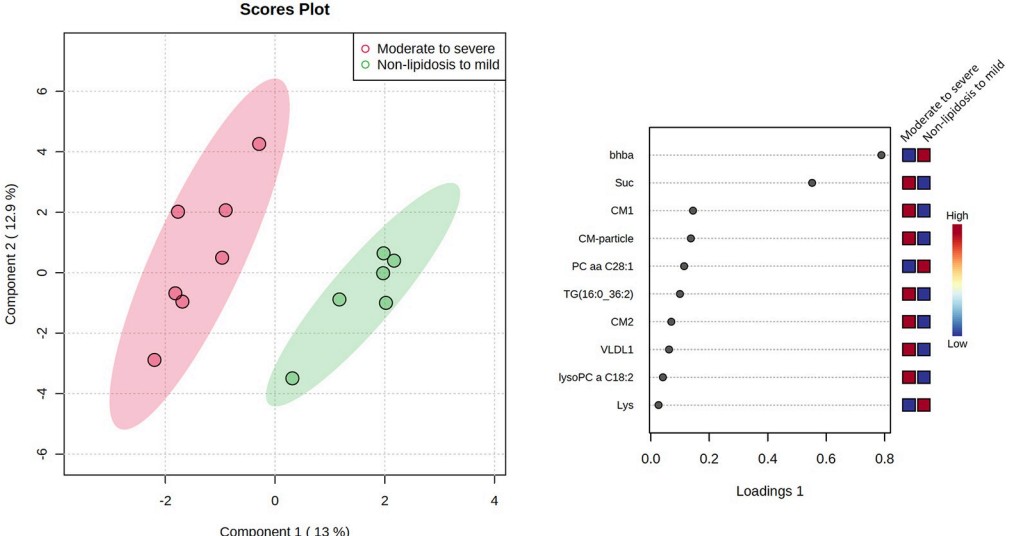

**Fig 5. Box plot with superimposed dot plot of beta-hydroxybutyric acid (umol/L) across two simplified classes of hepatic lipid changes (no-lipid to mild and moderate to severe) in bearded dragons based on an established grading system [30] evaluated by a t-test.** The dotted red line represents a cut-off value of 272 umol/L determined on receiver operating characteristic analysis.

promising and was an important variable in the sPLS-DA model. Hepatic lipidosis should be a differential diagnosis in bearded dragons with plasma BHBA lower than 272 umol/L or succinic acid higher than 13.7 umol/L. Gemfibrozil had a significant effect on succinic acid, by preventing its elevation in the treatment group, but was not found to be an effective treatment for hepatic lipidosis at the dose and frequency used, though animals in the moderate-to-severe class showed improvement in their biomarkers and reduced the amount of fat in their liver.

In this study, bearded dragons in the moderate-to-severe class weighed significantly more than those in the no-lipid-to-mild class. This weight distribution likely had to do with the age of the animals, as the majority of the bearded dragons that weighed less were smaller and approximately one year of age, while the larger bearded dragons that weighed more were between 2–4 years of age. Since age is a known risk factor for increasing histologic grade and class in the evaluation of hepatic lipidosis [1], this likely explains the distribution of disease. Only bearded dragons with spontaneous disease were included in this study and so randomization could not be performed to account for age and weight as confounder variables.

This study indicates that non-contrast CT accurately predicts the amount of fat in the liver of bearded dragons as there was only a single case that was misdiagnosed when correlated with histology. An increased sample size would allow for a more precise determination of the sensitivity of this diagnostic test. It is important to note that CT does not account for fibrosis, and attenuation should be interpreted in this context as the severity of cases with significant fibrosis may be under reported.

Although CT has not been used to extensively study reptile livers, it has been used to determine the mean liver density in apparently healthy Hermann's tortoises (*Testudo hermanni*) (50–70 HU), juvenile green sea turtles (*Chelodina mydas)* (60.09 +/- 5.3 HU), red eared sliders (55.78 +/- 11.66 HU), Blanding's turtles (*Emydoidea blandingii*) (97.5 HU +/- 9.6 HU), green iguanas (*Iguana iguana*) (77.30 +/- 6.2 HU), and free-ranging boa constrictors (*Boa constrictor*) (61.76 +/- 7.11 HU) [87–92]. In comparison, the mean (+/-SD) HU in this cohort of bearded dragons in the no-lipid-to-mild class was 44.03 +/- 11.21 HU. This value may be artificially lowered due to the inclusion of mild cases. Studies correlating histological hepatic fat

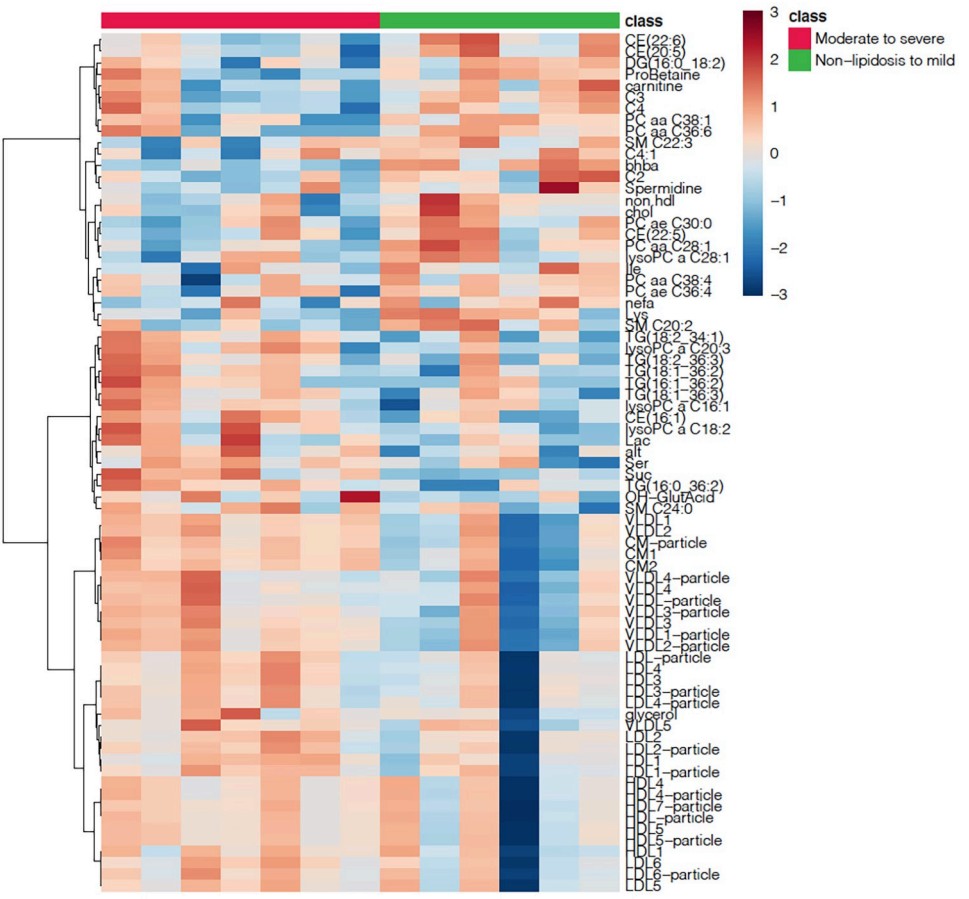

**Fig 6. Score plot of sparse partial least square- discrimination analysis (sPLS-DA) analysis of metabolomics, lipidomics, and biochemistry data between the 2 first principal components in bearded dragons with 2 simplified classes of hepatic lipid changes (no-lipid to mild and moderate to severe) in bearded dragons based on an established grading system [30].** The right panel shows the loadings plot for the first component showing the most important variables for classification.

content with hepatic density on CT in testudines concluded that liver density values greater than 55 +/- 11 HU were within normal limits, while values between 15–40 HU were indicative of hepatic lipidosis with values less than 20 HU, correlating with severe hepatic lipidosis [34]. The cut off for severe hepatic lipidosis in testudines was consistent with the cut off for the moderate-to-severe class in this study.

Subjective image evaluation, whether ultrasonographic or coelioscopic, was unreliable in determining the histologic class of the cases. The liver of healthy bearded dragons can have variable echogenicity on ultrasound, though the majority are hypoechoic to the fat pads and have a coarse echotexture [41, 42]. In healthy chameleons, the liver was found to be isoechoic to the adjacent fat bodies [93]. Though hyperechogenicity of the liver has been described for hepatic lipidosis in reptiles [94], the comparison of liver to fat pad echogenicity using pixel intensity was not a good classifier between classes. These results did trend in the right direction with the hepatic echogenicity increasing in comparison to the fat pads in the moderate-to-severe class and could be statistically significant with a larger sample size. Absolute echogenicity of the liver was not evaluated as ultrasonographers generally use a reference organ to determine echogenicity as this can change based on multiple factors [95].

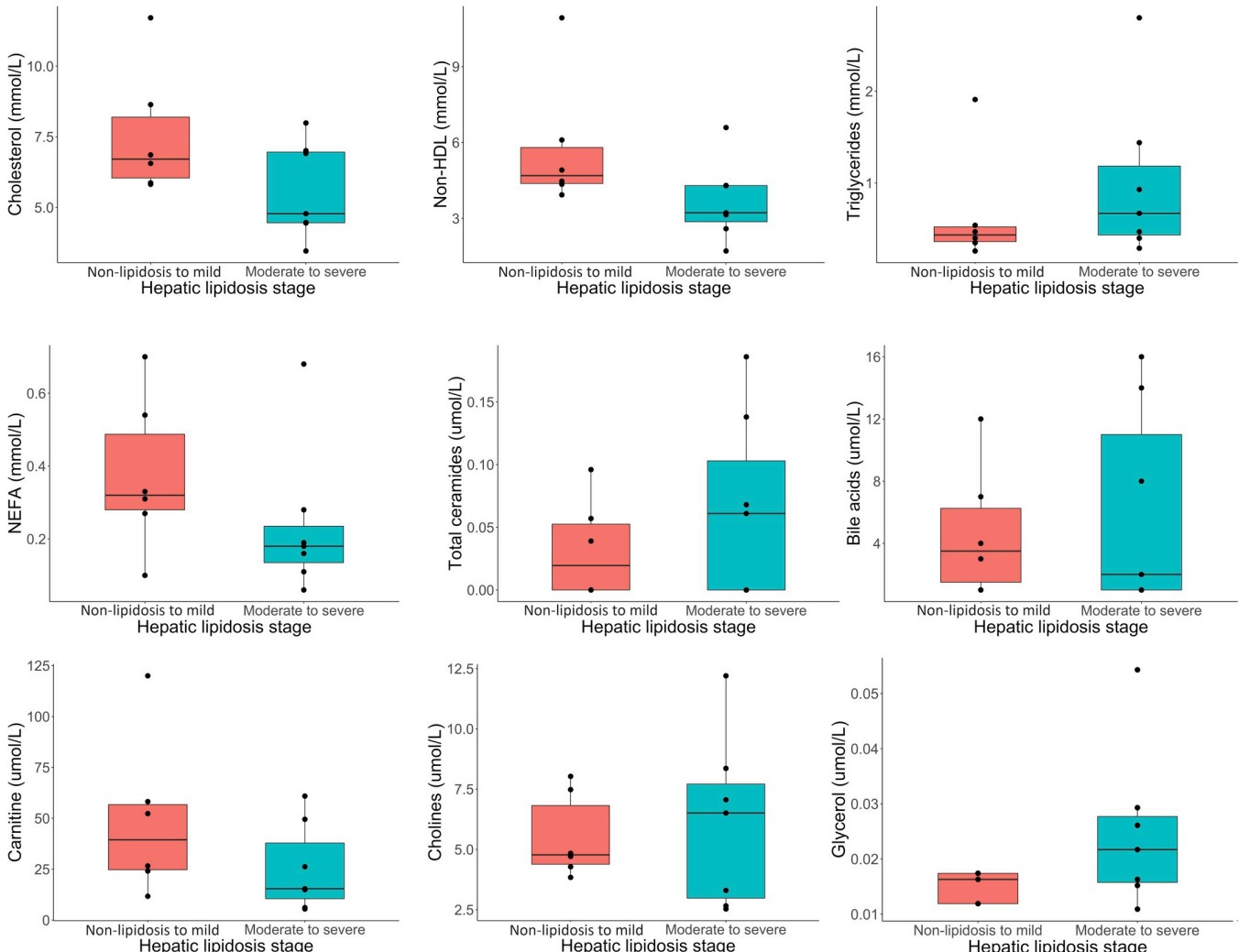

**Fig 7. Heatmap showing clustering of metabolites and biochemical analytes between simplified classes of hepatic lipid changes (no-lipid to mild and moderate to severe) in bearded dragons based on an established grading system [30].** A clustering dendrogram is also present on the left; the different bearded dragons are on the x-axis and the analytes on the y-axis. Only the 75 most important analytes based on their t-test p-values are displayed. It should be noted that most of these analytes did not show significant differences between histologic classes on univariate analysis with a false discovery rate of 0.05. Color coding represents fold changes on normalized plasma concentrations with an increasing depth of orange representing increase in analyte concentration and an increasing depth of blue representing decrease in analyte concentration.

Plasma biochemistry values are routinely evaluated as non-invasive biomarkers for hepatic lipidosis in human and veterinary medicine. However, these values are very insensitive to screen for this disorder and although liver analyte changes in association with hepatic lipidosis have been discussed in the literature [7, 9, 43, 91, 96–98], these values are generally within normal limits unless the disease is severe [5, 8, 13, 31, 99, 100]. In addition, reptile liver function tests have not been validated [5, 8] and pre and post prandial bile acid measurements in red-eared sliders (*Trachemys scripta elegans*) and green iguanas have produced inconsistent results [101, 102]. The results of this study confirmed that hepatocellular enzymes and bile acids did not significantly change in the moderate-to-severe class that was consistent with hepatic lipidosis, except for a non-significant increase in ALT seen on the heatmap. In bearded dragons, alterations in hepatic enzymes may not occur as cellular inflammation and necrosis do not seem to be a common feature of hepatic lipidosis [30]. Other tests available from most

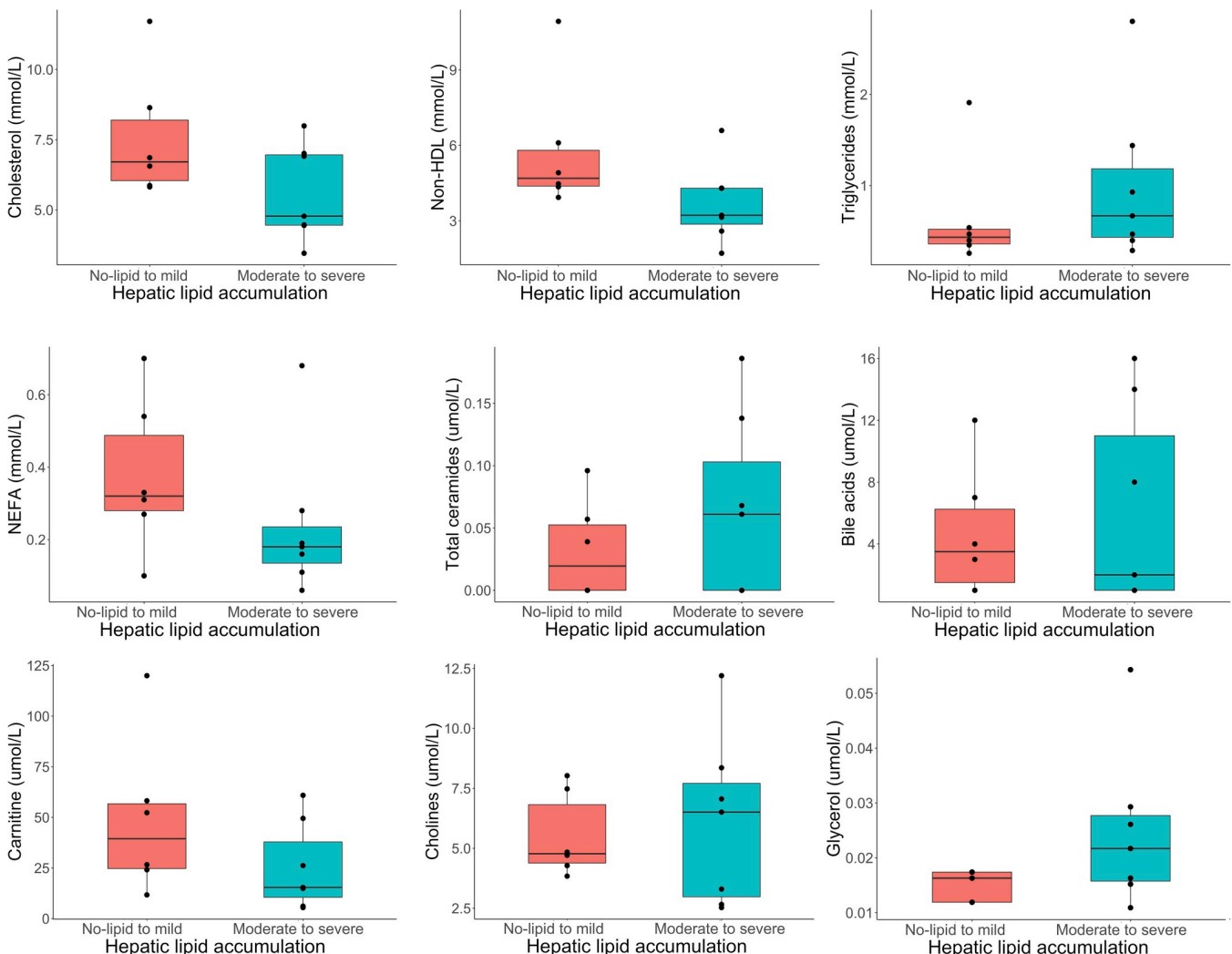

**Fig 8. Side-by-side box plots with superimposed dot plots of selected plasma variables (cholesterol [mmol/L], non-HDL [mol/L], NEFA [mmol/L], total ceramides [umol/L], carnitine [umol/L], and cholines [umol/l]) across two simplified classes of hepatic lipid changes (no-lipid to mild and moderate to severe) in bearded dragons based on an established grading system [30] evaluated by serial t-tests.** These variables were not statistically significant after correcting for false discovery, but interesting trends can be observed.

veterinary diagnostic laboratories in relation to lipid metabolism are seldom used in bearded dragons. These tests include NEFA, ketones including BHBA, and lipoproteins.

Of the analytes evaluated in this study, BHBA was found to be the best biomarker on multivariate analysis and the only statistically significant variable. This biomarker had a higher accuracy than CT to discriminate the no-lipid-to-mild class from moderate-to-severe class and is a fairly inexpensive test. Beta-hydroxybutyric acid is the main ketone acid produced by reptiles and is a by-product of fatty acid oxidation and ketogenesis [10]. Cases in the moderate-to-severe class generally have some component of microanatomy disruption, or hepatocellular swelling from triacylglycerol accumulation, and this disruption is suspected to be the inciting cause of metabolic and clinical disease [30]. The mechanisms of action that can be considered for reduced BHBA includes disruption of cellular function from triacylglycerol accumulation resulting in mitochondrial damage or toxicity [103] and impaired fatty acid oxidation and/or ketogenesis [104]. In humans, impaired ketogenesis is a hallmark of hepatic steatosis and the

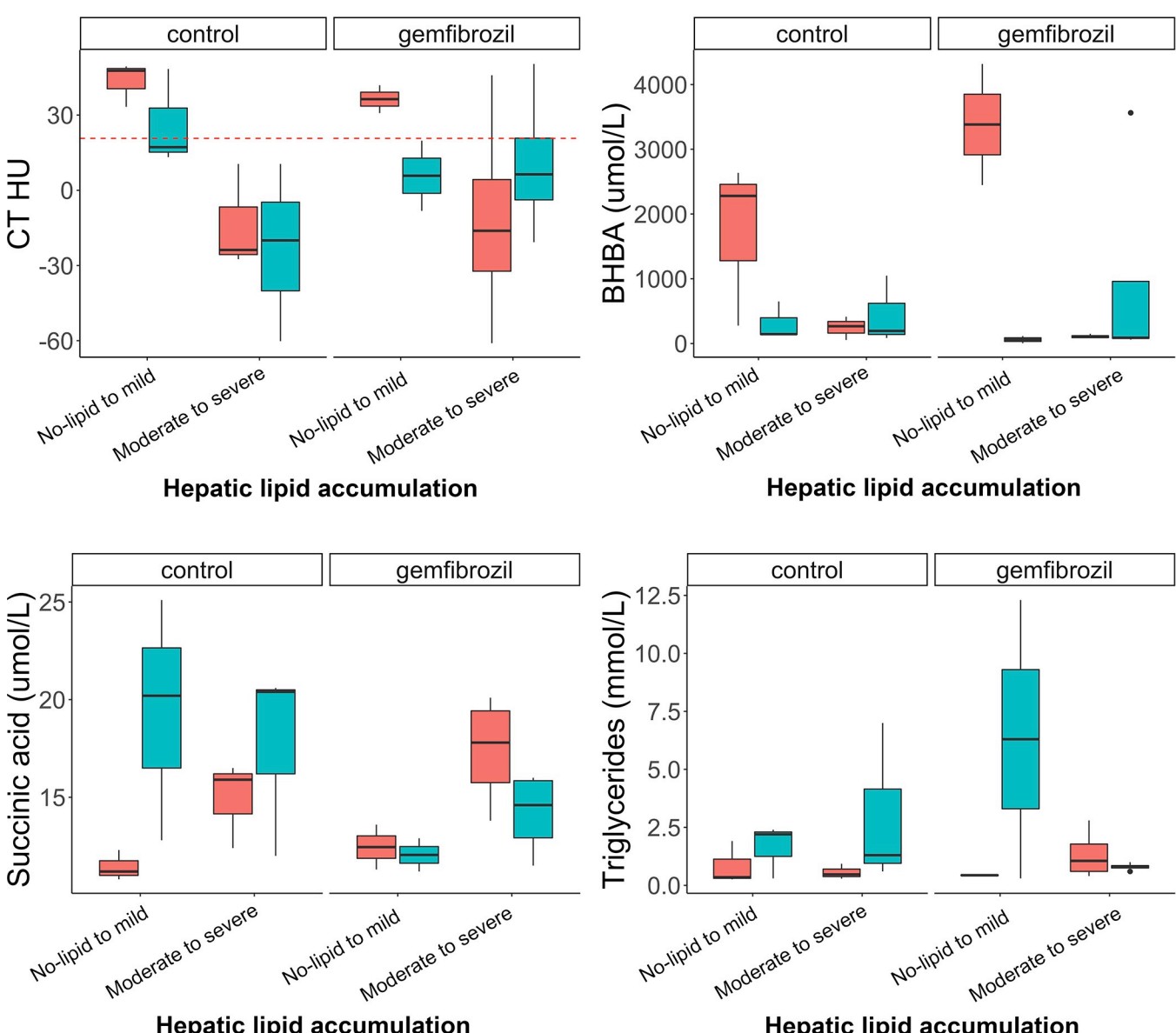

**Fig 9. Side-by-side box plots with superimposed dot plots of selected variables (CT [HU], BHBA [umol/L], succinic acid [umol/l], triglycerides [mmol/ L]) across two simplified classes of hepatic lipid changes (no-lipid to mild and moderate to severe), based on an established grading system [30], in bearded dragons with and without (control) gemfibrozil treatment evaluated by t-tests.** The red boxplots represent baseline values and the teal represents values after a 2-month treatment trial with gemfibrozil. The dotted red line on the CT HU side-by-side box plot represents a cut-off value of 21 HU to separate the classes of no-lipid to mild and moderate to severe as determined on receiver operating characteristic analysis.There was a significant effect of gemfibrozil on succinic acid. While succinic acid significantly increased over time in the control group, it did not with gemfibrozil treatment. The remainder of the variables were not statistically significant but did show interesting trends.

same may be true in bearded dragons [104]. Since BHBA may be affected by fasting, leading to increased fatty acid oxidation and ketogenesis, it should be performed on a fasted animal to be comparable to our results.

A biomarker of importance was detected in the metabolomics approach in this study- succinic acid. Based on the current study, succinic acid discriminates well between hepatic lipidosis classes in bearded dragons and was also the only parameter positively and significantly affected by the gemfibrozil therapy. Succinic acid is a key component of the tricarboxylic acid

cycle and is also the end-product of ω-oxidation of fatty acids which occurs in the endoplasmic reticulum. The tricarboxylic acid cycle has been shown to be altered in people with non-alcoholic steatosis and succinic acid seems to be the cycle intermediate that is most sensitive to buildup [105–107]. Mitochondrial damage form triacylglycerol accumulation may result in tricarboxylic acid cycle impairment in bearded dragons causing the build-up of succinic acid. In people, ω-oxidation is considered to be a rescue pathway for fatty acid oxidation when β-oxidation fails [108]. Therefore, it is possible that this pathway was upregulated in bearded dragons with mitochondrial dysfunction, which is plausible if a decrease in β-oxidation was to be confirmed as the cause for the decrease in BHBA. The decrease in ketogenesis may also be associated with an upregulation of the tricarboxylic acid cycle (TCA) cycle to metabolize the acetyl-CoA derived from β-oxidation that would normally serve to produce ketones, thus resulting in an increase in succinic acid [104]. Elevated levels of succinic acid with reduced levels of BHBA were found to be highly accurate in discriminating the moderate-to-severe class and should be considered biomarkers of choice for further evaluation in a new cohort of bearded dragons. Other potential biomarkers were detected on the sPLS-DA model and included TG 16:0_36:2, and PC 28:1. Various TGs and PCs have also been found as potential biomarkers in human studies [74, 76, 103].

While other metabolites were not specifically identified on the multivariate analysis as potential importance, their differential abundance across histologic class in bearded dragons may still shed some light on potential pathophysiological aspects.

Carnitine is the main transporter of fatty acids into the mitochondria in the form of acylcarnitines to undergo fatty acid oxidation [109]. Carnitine deficiency and alteration in the carnitine cycle has been suggested as contributing factors to hepatic lipidosis in cats [83, 109, 110]. Carnitine and several short-chained acylcarnitines were found to be in lower plasma concentrations in bearded dragons in the moderate-to-severe class. This disruption of the carnitine cycle may be both a cause (nutritional deficiency) or a consequence of an altered β-oxidation.

Trends were also seen in the glycerophospholipids and their precursors in the moderate-to-severe class. As the levels of PCs reduced, levels of LysoPCs (lysophosphatidylcholines) the monoacyl form of PCs, increased. Phosphatidylcholines, abundant phospholipids in mammals and reptiles [111, 112], play an essential role in the secretion of lipoproteins [113]. These glycerophospholipids have been implicated in the pathogenesis of hepatic lipidosis in mammals as reduced levels result in impaired VLDL secretion from the liver resulting in hepatic accumulation of triacylgcerols [74, 114, 115]. In addition, low concentrations of PCs are known to upregulate hepatic lipogenesis, furthering lipid accumulation in these hepatocytes [82, 116]. The accumulation of LysoPCs is possibly due to reduction of their acylation thereby reducing the amount of PCs available. Increased levels of LysoPCs downregulate genes involved in fatty acid oxidation and can compromise mitochondrial integrity [117].

The triglycerides with 52 carbons (C52) in the acyl chains (especially TG 16:0_36:2) tended to be higher in bearded dragons in the moderate-to-severe class. In people, an increase in saturated triglycerides is a marker of hepatic lipidosis and the elevation is thought to occur through de novo lipogenesis, as the primary products of this process are saturated fatty acids [118]. Palmitic acid, in particular, is a potent mediator of lipotoxicity [119]. These saturated triacylglycerols are known to have a negative effect on mitochondrial metabolism [118, 120]. A similar process with similar implications could be occurring in the bearded dragons in this study. However, fatty acid metabolism and regulation is likely different in reptiles and the composition of fatty acids is temperature dependent with a tendency to have more saturated fatty acids in the liver as the thermal gradient increases [6, 26, 27, 121]. Serum or plasma triglycerides are reported to be elevated in reptiles during pre-brumation, vitellogenesis, follicular stasis, and hepatic lipidosis [3, 5, 8, 28]. Studies on Whiptail lizards (*Cnemidophorus tigris*) prior to

brumation demonstrated a different fatty acid profile to the bearded dragons with hepatic lipidosis in this study, as the main hepatic fatty acids were unsaturated fatty acids -oleic and linolenic acid [25]. This may be a vital difference between physiological and pathological fat accumulation in reptiles and further investigation is warranted. Another factor to consider in evaluation of triglycerides is the process of vitellogenesis in female bearded dragons. Only one female bearded dragon in the moderate-to-severe class had active follicles during diagnostic testing and so this pathophysiology was not considered to be significant in this study.

Ceramides are simple sphingolipids that are important for the structural component of the cell membrane [122]. Plasma elevations in people have been correlated with progression of hepatic steatosis which has resulted in the development of routine ceramides testing [20, 81]. Ceramides are thought to decrease insulin sensitivity, increase the production of cytokines involved in inflammation, and increase oxidative stress and mitochondrial dysfunction [123]. Ceramides were not elevated in the more severely affected bearded dragons which may be a reason why inflammation does not seem to be an obvious component of their hepatic lipidosis [30].

On various analyses of bearded dragons in the moderate-to-severe class, triglycerides rich in lipoproteins had a tendency to be higher than the no-lipid-to-mild class. These findings are likely linked to higher triglyceride levels in the plasma from dyslipidemia and hepatic lipidosis as demonstrated in mammals and birds [124–127].

Free glycerol increased in bearded dragons in the moderate-to-severe class and was likely due to reduced gluconeogenesis as suggested in people with non-alcoholic fatty liver disease [75].

Lastly, though not statistically significant, markers of hepatic damage and dysfunction, including ALT and lactic acid, had an upwards trend in bearded dragons in the moderate-to-severe class. These bearded dragons likely had a component of hepatocyte damage and dysfunction with disease progression, though it seems progression would have to be significant prior to a marked elevation. The excluded bearded dragon with hepatic amyloidosis had severely elevated ALT activities with other liver parameters being normal. It is noteworthy that bile acids were not useful to diagnose hepatic lipidosis in bearded dragons, despite being advocated as a sensitive and specific liver parameter on biochemistry profiles [128]. Further investigation into the usefulness of plasma ALT in bearded dragon hepatic diseases is warranted.

The abnormal fatty acid/ketone metabolism occurring in bearded dragons has a different pathogenesis than what is reported in cats where an acute syndrome occurs secondary to anorexia, resulting in peripheral lipolysis and a high hepatic influx of NEFA which exceeds the liver's maximum rate of fatty-acid β-oxidation [129–131]. Increased hepatic β-oxidation results in a marked elevation of BHBA, which has been suggested as a feline hepatic lipidosis biomarker [130, 132]. In bearded dragons, this process does not seem to be related to anorexia. Instead, it is likely that there is a physiological but progressive accumulation of triacylglycerols in the liver for various reasons such as preparation for vitellogenesis or brumation [25], or in response to nutritional factors such as excess carbohydrates and fat. High sugar diets, even if isocaloric, have been shown to increase the levels of triacylglycerol, acylcarnitine, and downregulate genes involved in fatty acid oxidation in rats [83]. If bearded dragons fail to metabolize these stored triacylglycerols from lack of reproduction, brumation [133], or activity, a pathological process may begin to occur. Hepatic lipidosis in bearded dragons seems to be associated with an increase in triglycerides, triglyceride-rich lipoproteins (including chylomicrons and large VLDLs), free glycerol, and lysoPC. Carnitine, short chain acylcarnitines, PC, NEFA, BHBA, lysine, total cholesterol and non-HDL trend downwards. Together these changes indicate that there is an alteration of fatty acid metabolism with a decrease in lipolysis markers, decreased hepatic uptake of dietary lipoproteins, and a decrease in metabolites that transport

fatty acids to the mitochondria, resulting in a decrease of β-oxidation and/or alteration of acetyl-CoA disposal pathways. Inhibition of this pathway results in a buildup of glycerophospholipids in the plasma and possible upregulation of other pathways such as fatty acid ω-oxidation and the TCA cycle. As liver function decreases, markers of liver injury such as ALT and lactic acid tend to increase.

Gemfibrozil, a drug which strongly promotes fatty acid oxidation in the liver, was selected to help reverse the metabolic state of hepatic lipidosis, ultimately leading to a decrease in hepatic triacylglycerol concentration [45–48]. This medication has been studied in avian species demonstrating significant effects on triacylglycerol concentrations and has been shown to decrease the average lipoprotein density [10, 50, 134]. Pharmacokinetic data in bearded dragons is lacking and so a dose was extrapolated based on allometric scaling [69]. While safety of gemfibrozil is unknown in bearded dragons, it is considered to be an extremely safe drug with a $LD_{50}$ of 2218 mg/kg in mice and overdoses of more than 100 times the therapeutic dose not resulting in lasting effects in humans [135].

In the pharmacodynamic study of gemfibrozil's effect on hepatic lipidosis, there was a significant effect of treatment as succinic acid significantly increased in the control group over time while it did not in the treatment group. Even with this effect, gemfibrozil at the dose used, did not prevent the progression of hepatic lipid accumulation in bearded dragons in the no-lipid-to-mild class. However, these results were likely confounded by the fact that this class contained all the younger animals who likely became mature adults over the 4.5 months between the initial diagnostics and post-trial diagnostics. This made evaluation of gemfibrozil's efficacy difficult as biomarkers were compared to baseline and these bearded dragons had reduction of BHBA and an increase of triglycerides following sexual maturity.

There was a tendency in animals of the moderate-to-severe class to improve their biomarker profile and decrease the amount of fat in the liver in the treatment group while the matched control group had increased fat in the liver and worsening of some biomarkers (increased succinic acid and triacylglycerols). The trend of an increase in BHBA and stabilization of triglycerides suggests that gemfibrozil may have promoted fatty acid oxidation and ketogenesis in the liver, which downregulated the disposal of acetyl-CoA through the TCA cycle or ω-oxidation. This was evident by increased hepatic attenuation and reduced fat percentage on digital image analysis suggesting reduction of hepatic triacylglycerol. Histological evidence of improvement was only seen in half of the gemfibrozil cases. This is likely due to the fact that more significant changes are required prior to a final score improvement using the grading scheme. Further studies are required to determine further efficacy with a larger sample size, better control of variability while matching for age, or a higher dose or treatment length as reptiles are known to have a slower metabolism [136].

Various limitations were present in this research. A homogenous population was not present in this study since 5 smaller bearded dragons were acquired from a breeder. This produced an uneven age and weight distribution across the two simplified classes which may have added confounding variables. However, this strategy was elected to obtain animals with and without spontaneous hepatic lipidosis to perform the clinical trial and it was difficult to obtain adult bearded dragons in another manner.

Dietary and husbandry recommendations from literature were used to determine the husbandry plan for the bearded dragons in this study. However, many of these recommendations are not evidence based and lack clarity [137]. The progression of histologic changes in this study population suggests further evidence-based recommendations on captive husbandry in bearded dragons is needed. Confounding factors to consider for the population include the lack of exercise during the study period and lack of brumation. To prevent the spread of infectious diseases, the bearded dragons were not provided communal areas to exercise. In

addition, the physiological level of fat in the liver is still unknown in bearded dragons and so this progression of lipid accumulation may be maturation of these bearded dragons from sub-adult to adult.

The site of hepatic biopsies was not precisely defined though all were taken from the margin of the left liver lobe. Therefore, a precise correlation between the histology sample and the attenuation measurements from the ROI could not be completed. Though previous evaluation of 252 histology slides indicated that the majority of hepatic lipidosis cases are diffuse and pan-lobular, focal infiltration could have been missed due to the small size of the biopsy samples [30].

Due to the exploratory nature of the metabolomics and lipidomics, a large number of variables were analyzed on a small sample size of individuals, which significantly reduced the statistical power of this study. These results should therefore, be considered more useful for generating hypotheses to be further tested in prospective observational and experimental studies using higher sample sizes. Likewise, most of this research had a hypothesis generating focus rather than a hypothesis driven focus.

## Conclusions

In conclusion, this pilot study lays the groundwork for further research on biomarker discovery, pathophysiology of hepatic lipidosis, and pharmacologic intervention of this very common disease in pet bearded dragons. Based on these findings, the authors considered the moderate-to-severe class to be consistent with the diagnosis of hepatic lipidosis. Follow up prospective studies controlling for age, sex, and weight, or experimental models with induced disease, are required to confirm and validate biomarkers in the diagnosis of hepatic lipidosis as well as their response to other hepatic disorders. Computed tomography is a reliable and objective measure of lipid vacuolization, though biopsies are still required for assessment of fibrosis. However, it can be used in future biomarker studies as a non-invasive confirmatory test for hepatocyte vacuolization and to monitor the progression or improvement of this disease process. Beta-hydroxybutyric acid and succinic acid are both promising biomarkers in the diagnosis of hepatic lipidosis, but more data is needed to confirm their usefulness. In clinical practice, bearded dragons need to be fasted for 24 hours for these biomarkers to be comparable to our results as metabolite profiles are likely different in a fed state. In addition, these profiles may alter in anorexic animals who are in a catabolic state or with different environmental parameters such as their thermal gradient. None of the bearded dragons in this study were anorexic. While gemfibrozil only showed significant efficacy in one biomarker, many of the other biomarkers were trending in the right direction. Hepatic lipidosis is a difficult disease to treat and further investigations on this class of drugs are warranted with these encouraging, albeit modest, results. Pharmacokinetic data will also be useful to guide further pharmacological studies on fibrate therapy in reptiles.

## Supporting information

**S1 Table. Histological grading system and severity classification for changes associated with diffuse and panlobular hepatic lipid accumulation in bearded dragons (*Pogona vitti-ceps).* **Each category is given a score from 0–4 based on the percentage of change in the histologic section of evaluated liver. Subsequently, a cumulative grade is used to determine the severity classification. Estimations using 400X magnification.
(DOCX)

## Acknowledgments

Exoterra Sunray lamps were provided for free by Hagen Avicultural Research Institute.

We also thank Reptilia, Vaughan and Cheryl Sheridan for acquiring and housing bearded dragons prior to this research.

## Author Contributions

**Conceptualization:** Trinita K. Barboza, Alex zur Linden, Hugues Beaufrère.

**Data curation:** Trinita K. Barboza, Hugues Beaufrère.

**Formal analysis:** Trinita K. Barboza, Hugues Beaufrère.

**Funding acquisition:** Trinita K. Barboza, Leonardo Susta, Alex zur Linden, Hugues Beaufrère.

**Investigation:** Trinita K. Barboza, Leonardo Susta, Alex zur Linden, Sara Gardhouse, Hugues Beaufrère.

**Methodology:** Hugues Beaufrère.

**Project administration:** Trinita K. Barboza, Hugues Beaufrère.

**Supervision:** Trinita K. Barboza, Hugues Beaufrère.

**Visualization:** Trinita K. Barboza, Hugues Beaufrère.

**Writing – original draft:** Trinita K. Barboza, Hugues Beaufrère.

**Writing – review & editing:** Leonardo Susta, Alex zur Linden, Sara Gardhouse, Hugues Beaufrère.

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
