## [Decision Letter · Decision Letter 0]

11 Apr 2022

PONE-D-22-04396Association of plasma metabolites and diagnostic imaging findings with hepatic lipidosis in bearded dragons (Pogona vitticeps) and effects of gemfibrozil therapyPLOS ONE

Dear Dr. Beaufrere,

Thank you for submitting your manuscript to PLOS ONE. After careful consideration, we feel that it has merit but does not fully meet PLOS ONE’s publication criteria as it currently stands. Therefore, we invite you to submit a revised version of the manuscript that addresses the points raised during the review process.

We look forward to receiving your revised manuscript.

Kind regards,

Nobuyuki Takahashi, Ph.D.

Academic Editor

PLOS ONE

Journal Requirements:

2. We noted in your submission details that a portion of your manuscript may have been presented or published elsewhere. 

(Results were published in a DVSc thesis at the University of Guelph:

https://atrium.lib.uoguelph.ca/xmlui/handle/10214/26230)

Please clarify whether this publication was peer-reviewed and formally published. If this work was previously peer-reviewed and published, in the cover letter please provide the reason that this work does not constitute dual publication and should be included in the current manuscript.

Additional Editor Comments:

So sorry for the delay of reviewing this manuscript. It was difficult to find reviewers.

Reviewers' comments:

Reviewer's Responses to Questions

**Comments to the Author**

1. Is the manuscript technically sound, and do the data support the conclusions?

Reviewer #1: Yes

Reviewer #2: Yes

2. Has the statistical analysis been performed appropriately and rigorously? 

Reviewer #1: Yes

Reviewer #2: I Don't Know

3. Have the authors made all data underlying the findings in their manuscript fully available?

Reviewer #1: Yes

Reviewer #2: Yes

4. Is the manuscript presented in an intelligible fashion and written in standard English?

Reviewer #1: Yes

Reviewer #2: Yes

5. Review Comments to the Author

Reviewer #1: Well-done. Important manuscript focused on an important captive species. I enjoyed reading how the authors worked through the problem of hepatic lipidosis by pursuing cutting edge methods to develop a better understanding of this important disease in this species. Limitations well thought out and described. I classify this as a major revisions only because there were more than a handful of comments, but most are suggestions for grammar, etc. Hope these veterinary scientists can continue to pursue this research.

Reviewer #2: Authors demonstrated here that BHBA and succinate were very good candidates as biomarkers for abnormalities in hepatic lipid metabolism of bearded dragons and that gemfibrozil was good for the treatment of fatty liver in bearded dragons. These findings are very important and valuable for understanding bearded dragons' abnormality in lipid metabolism. However, several points in examination of gemfibrozil's effects should be confirmed before publish as the followings, although finding biomarkers is good. In addition, authors should modify this manuscript according to basic format as a manuscript submitted to PONE, because this manuscript was not described in a standard format.

Majors:

1) Authors should show CT data of gemfibrozil-treated bearded dragons. The data would be useful for diagnosis and treatment of fatty liver.

2) Serum parameters of gemfibrozil-treated bearded dragons should be shown. An important effect of gemfibrozil is to decrease serum TG levels. Therefore, data on serum parameters such as TG levels are essential for evaluating the effects of gemfibrozil. If possible, the time course of TG levels during gemfibrozil administration should be indicated.

Minors:

1) The "Materials and Methods" section should be rearranged. In the "Results" section, animal information should not be described and only explanation of data should be described. The animal information should be moved to the "Materials and Methods." In the "Materials and Methods", "Inclusion criteria, animals, exclusion criteria" is too long and broken into small items including the animal information described in the "Results" now.

2) Figure legends should be revised. Normally, figure legends should contain information on the display of data. The figure legends in this manuscript is too little information and difficult to understand. It is necessary to describe in Figure legends what kind of values, what kind of graphs, how to do statistical analysis, etc. Please check other articles on PONE carefully.

6. PLOS authors have the option to publish the peer review history of their article (what does this mean?). If published, this will include your full peer review and any attached files.

Reviewer #1: **Yes: **Mark A. Mitchell

Reviewer #2: No

---

## [Author Response · Author response to Decision Letter 0]

17 Jun 2022

¬¬Dear PLOS ONE reviewers and editors, 

Thank you for reviewing our manuscript. The authors accepted all edits throughout the manuscript and added missing manufacturer information and clarified the anesthesia protocol. 

Journal Requirements

We noted in your submission details that a portion of your manuscript may have been presented or published elsewhere. 

(Results were published in a DVSc thesis at the University of Guelph:

https://atrium.lib.uoguelph.ca/xmlui/handle/10214/26230)

Please clarify whether this publication was peer-reviewed and formally published. If this work was previously peer-reviewed and published, in the cover letter please provide the reason that this work does not constitute dual publication and should be included in the current manuscript.

Response: This manuscript has not been peer-reviewed and formally published. It is included in a Doctor of Veterinary Science (DVSc) thesis that was successfully defended and is available online for review. 

Reviewer 1:

Reviewer comment line 503 “small sample, low power. Risk of type II error?”

Response: Authors agree that the sample size was low in this pilot study and so acknowledged this in the discussion. Line 716 modified to say: “These results did trend in the right direction with the hepatic echogenicity increasing in comparison to the fat pads in the moderate-to-severe class and could be statistically significant with a larger sample size.”

Reviewer comment line 659 “would suggest adding these references here”

Response: Additional references were not visible on the PDF for authors to add. 

Reviewer 2: 

Authors demonstrated here that BHBA and succinate were very good candidates as biomarkers for abnormalities in hepatic lipid metabolism of bearded dragons and that gemfibrozil was good for the treatment of fatty liver in bearded dragons. These findings are very important and valuable for understanding bearded dragons' abnormality in lipid metabolism. However, several points in examination of gemfibrozil's effects should be confirmed before publish as the followings, although finding biomarkers is good. In addition, authors should modify this manuscript according to basic format as a manuscript submitted to PONE, because this manuscript was not described in a standard format.

Response: Authors changed the format of the supporting information file from docx to pdf, and added a conclusions section. 

Majors:

1) Authors should show CT data of gemfibrozil-treated bearded dragons. The data would be useful for diagnosis and treatment of fatty liver.

Response: All data is available in the repository link in the results section of the manuscript. In addition, the CT HU data are presented in the plots and the text. If you meant that more CT images should be present, this is certainly something we could add but representative CT images were already displayed.

2) Serum parameters of gemfibrozil-treated bearded dragons should be shown. An important effect of gemfibrozil is to decrease serum TG levels. Therefore, data on serum parameters such as TG levels are essential for evaluating the effects of gemfibrozil. If possible, the time course of TG levels during gemfibrozil administration should be indicated.

Response: All data is available in the repository link in the results section of the manuscript. Triglyceride levels were only measured once prior to gemfibrozil therapy and once after completion of therapy. Figure 9 shows box plots of these baseline and post treatment levels in the treatment and control group. 

Minors:

3) The "Materials and Methods" section should be rearranged. In the "Results" section, animal information should not be described and only explanation of data should be described. The animal information should be moved to the "Materials and Methods." In the "Materials and Methods", "Inclusion criteria, animals, exclusion criteria" is too long and broken into small items including the animal information described in the "Results" now.

Response: The “Animals” section was removed from the results and incorporated into the materials and methods under “Coelioscopic guided liver biopsy and histology” as it describes the weight distributions of the different histological classes. “Inclusion criteria, animals, exclusion criteria” was shortened by removing sections on bearded dragon initial admission and parasite screening and cricket gut-loading.

We would like to keep some of the information on husbandry as it is often asked by reviewers or needed by readers to ensure biochemistry changes and other physiological parameters have not been confounded by an inappropriate husbandry (mainly environmental parameters) and diet. This information could also be helpful to researchers interested in keeping bearded dragons as laboratory animals.

4) Figure legends should be revised. Normally, figure legends should contain information on the display of data. The figure legends in this manuscript is too little information and difficult to understand. It is necessary to describe in Figure legends what kind of values, what kind of graphs, how to do statistical analysis, etc. Please check other articles on PONE carefully.

Response: Authors modified figure legends to describe the kind of values, kind of graphs, and statistical analysis. Please see modifications made to the manuscript.

---

## [Decision Letter · Decision Letter 1]

22 Aug 2022

Association of plasma metabolites and diagnostic imaging findings with hepatic lipidosis in bearded dragons (Pogona vitticeps) and effects of gemfibrozil therapy

PONE-D-22-04396R1

Dear Dr. Beaufrere,

We’re pleased to inform you that your manuscript has been judged scientifically suitable for publication and will be formally accepted for publication once it meets all outstanding technical requirements.

Kind regards,

Nobuyuki Takahashi, Ph.D.

Academic Editor

PLOS ONE

Additional Editor Comments (optional):

Reviewers' comments:

Reviewer's Responses to Questions

**Comments to the Author**

1. If the authors have adequately addressed your comments raised in a previous round of review and you feel that this manuscript is now acceptable for publication, you may indicate that here to bypass the “Comments to the Author” section, enter your conflict of interest statement in the “Confidential to Editor” section, and submit your "Accept" recommendation.

Reviewer #1: All comments have been addressed

Reviewer #2: All comments have been addressed

2. Is the manuscript technically sound, and do the data support the conclusions?

Reviewer #1: Yes

Reviewer #2: Yes

3. Has the statistical analysis been performed appropriately and rigorously? 

Reviewer #1: Yes

Reviewer #2: Yes

4. Have the authors made all data underlying the findings in their manuscript fully available?

Reviewer #1: Yes

Reviewer #2: Yes

5. Is the manuscript presented in an intelligible fashion and written in standard English?

Reviewer #1: Yes

Reviewer #2: Yes

6. Review Comments to the Author

Reviewer #1: Thank you for addressing the comments. The article provides additional insight into an important disease in bearded dragons (and captive reptiles in general).

Reviewer #2: (No Response)

7. PLOS authors have the option to publish the peer review history of their article (what does this mean?). If published, this will include your full peer review and any attached files.

Reviewer #1: **Yes: **Mark A. Mitchell

Reviewer #2: No

---

## [Editor Report · Acceptance letter]

24 Aug 2022

PONE-D-22-04396R1 

Association of plasma metabolites and diagnostic imaging findings with hepatic lipidosis in bearded dragons (*Pogona vitticeps*) and effects of gemfibrozil therapy 

Dear Dr. Beaufrère:

I'm pleased to inform you that your manuscript has been deemed suitable for publication in PLOS ONE. Congratulations! Your manuscript is now with our production department. 

Kind regards, 

on behalf of

Dr. Nobuyuki Takahashi 

Academic Editor

PLOS ONE